# Angiopoietin-2 and the Vascular Endothelial Growth Factor Promote Migration and Invasion in Hepatocellular Carcinoma- and Intrahepatic Cholangiocarcinoma-Derived Spheroids

**DOI:** 10.3390/biomedicines12010087

**Published:** 2023-12-30

**Authors:** Adriana Romanzi, Fabiola Milosa, Gemma Marcelli, Rosina Maria Critelli, Simone Lasagni, Isabella Gigante, Francesco Dituri, Filippo Schepis, Massimiliano Cadamuro, Gianluigi Giannelli, Luca Fabris, Erica Villa

**Affiliations:** 1Department of Biomedical, Metabolic and Neural Sciences, Clinical and Experimental Medicine Program, University of Modena and Reggio Emilia, 41125 Modena, Italy; adriana.romanzi@unimore.it (A.R.); simone.lasagni@unimore.it (S.L.); 2Chimomo Department, Gastroenterology Unit, University of Modena and Reggio Emilia, 41125 Modena, Italy; fabiola.milosa@unimore.it (F.M.); gemma.marcelli@unimore.it (G.M.); rosinamaria.critelli@unimore.it (R.M.C.); filippo.schepis@unimore.it (F.S.); 3National Institute of Gastroenterology IRCCS “Saverio de Bellis”, Research Hospital, 70013 Castellana Grotte, Italy; isabella.gigante@irccsdebellis.it (I.G.); francesco.dituri@irccsdebellis.it (F.D.); gianluigi.giannelli@irccsdebellis.it (G.G.); 4Department of Molecular Medicine, School of Medicine, University of Padua, 35121 Padua, Italy; massimiliano.cadamuro@gmail.com (M.C.); luca.fabris@unipd.it (L.F.)

**Keywords:** proangiogenic factors, VEGF, Angiopoietin-2, 3D cancer model, migration, invasiveness, epithelial–mesenchymal transition, Trebananib, Bevacizumab

## Abstract

Aggressive hepatocellular carcinoma (HCC) overexpressing Angiopoietin-2 (ANG-2) (a protein linked with angiogenesis, proliferation, and epithelial–mesenchymal transition (EMT)), shares 95% of up-regulated genes and a similar poor prognosis with the proliferative subgroup of intrahepatic cholangiocarcinoma (iCCA). We analyzed the pro-invasive effect of ANG-2 and its regulator vascular endothelial growth factor (VEGF) on HCC and CCA spheroids to uncover posUsible common ways of response. Four cell lines were used: Hep3B and HepG2 (HCC), HuCC-T1 (iCCA), and EGI-1 (extrahepatic CCA). We treated the spheroids with recombinant human (rh) ANG-2 and/or VEGF and then observed the changes at the baseline, after 24 h, and again after 48 h. Proangiogenic stimuli increased migration and invasion capability in HCC- and iCCA-derived spheroids and were associated with a modification in EMT phenotypic markers (a decrease in E-cadherin and an increase in N-cadherin and Vimentin), especially at the migration front. Inhibitors targeting ANG-2 (Trebananib) and the VEGF (Bevacizumab) effectively blocked the migration ability of spheroids that had been stimulated with rh-ANG-2 and rh-VEGF. Overall, our findings highlight the critical role played by ANG-2 and the VEGF in enhancing the ability of HCC- and iCCA-derived spheroids to migrate and invade, which are key processes in cancer progression.

## 1. Introduction

Hepatocellular carcinoma (HCC) and intrahepatic cholangiocarcinoma (iCCA) are the primary malignancies of the liver. Traditionally, HCC is strongly associated with chronic liver disease-induced inflammation, while iCCA has a less pronounced connection to inflammatory processes [1]. The prognostic outlook and therapeutic resistance differ markedly between HCC and iCCA; iCCA typically has poorer outcomes, with a 5-year survival rate not exceeding 50% post-liver transplantation. In contrast, HCC has approximately a 30% cure rate with current therapies, and liver transplantation offers a 60–70% 5-year survival rate [2,3]. These disparities are less distinct when comparing the proliferative subtype of iCCA, as described by Sia et al. [4], with an aggressive HCC subgroup characterized by a five-gene ‘neoangiogenic’ transcriptomic signature (TS) [5,6]. Notably, approximately 20% of the iCCA biopsies we analyzed for gene expression exhibited the TS (Appendix A). Angiopoietin-2 (*ANG-2*), a key upregulated gene in the neoangiogenic TS, is primarily involved in physiological vascular remodeling [7,8] and the vascular wound healing processes [9]. Elevated levels of the protein ANG-2 have been observed at endothelial sites in various pathological states. This elevation leads to an increase in vascular permeability—a key factor in disease progression [9]. Particularly in cancer, the overexpression of ANG-2 triggers the process of neoangiogenesis. This is the creation of new, poorly structured, and unstable blood vessels, which, in turn, can accelerate tumor growth [10,11]. It is noteworthy that researchers have found a correlation between the levels of ANG-2 and the development of tumors. This correlation is particularly strong in cases of HCC, especially its more aggressive forms [5,12]. Similarly, in CCA, ANG-2 has been identified as a significant biomarker that can aid in diagnosis and predict outcomes [13]. The vascular endothelial growth factor, commonly known as VEGF, is a protein that modifies the activity of ANG-2. This interaction occurs in normal bodily processes, as well as in the development of tumors [14]. In normal physiological conditions, the VEGF helps in the growth of endothelial cells. However, in cancer, especially in liver cancer HCC and CCA, high levels of the VEGF are linked to a lower survival rate [15,16]. Research has shown that ANG-2 works together with the VEGF to enhance blood vessel formation and the growth of tumors, leading to larger tumors and an increased risk of the cancer spreading [14,17,18]. Interestingly, inhibiting the action of the VEGF can reduce these harmful effects, particularly in HCC. This suggests that it is the combined effect of ANG-2 and the VEGF that significantly contributes to tumor growth and the development of new blood vessels [19]. Despite the role of ANG-2 and the VEGF in the clinical setting of HCC and CCA being well described, the direct effect of ANG-2 and the VEGF, alone or together, on liver malignant cells in vitro has not been well elucidated. Most research has focused on how endothelial cells affect angiogenesis in cancer. However, fewer studies have explored their direct impact on cancer cells. For instance, ANG-2 has been found to increase the growth of tumor cells in pituitary neuroendocrine tumors [20]. Similarly, stimulating cells with a specific form of recombinant human ANG-2 (rh-ANG-2) can enhance the invasive capabilities of glioma cells [21]. In the context of colorectal cancer and HCC, the VEGF plays a crucial role in cell migration and invasion. It is also important for the proliferation of cancer cells in mouse epithelial tumors [22,23]. We demonstrated a direct role of these proangiogenic factors on HCC cells and, for the first time to our knowledge, on iCCA cells. Building on previous research that highlights the synergistic effects of ANG-2 and the VEGF, we explored whether combining these factors would have a more significant impact than using them separately, even in in vitro conditions.

In various cancers, including liver cancer, researchers have found a connection between angiogenic stimuli and the process of epithelial–mesenchymal transition (EMT). Studies by Dong et al. [24], Ribatti et al. [25], and Giannelli et al. [26] have shown that EMT plays a key role in driving tumor progression by increasing invasion and metastasis. In some in vitro cancer models, such as melanoma, cervical carcinoma, and breast cancer, the role of EMT markers (i.e., N-cadherin, Vimentin, Snail, and Twist) as promoters of cell motility and increased resistance to an anti-cancer treatment has been described [27,28]. Also, in HCC and CCA, recent research has emphasized the clinical importance of EMT. This includes its potential as a target for cancer therapy and its use as a prognostic indicator, where the presence of certain EMT markers can provide insights into the likely course of the disease [29,30].

The growing significance of 3D models in cancer research, particularly for liver cancer, led us to investigate the effects of ANG-2 and the VEGF on cancer spheroids. We focused especially on the EMT process. Our aim was to uncover possible similarities in the responses of HCC and iCCA to these proangiogenic stimuli, which might highlight key pathways in cancer progression.

## 2. Materials and Methods

### 2.1. Cell Cultures

In this study, we used four different cell lines to understand liver and biliary tract cancers better. This included two cell lines from HCC (HepG2 and Hep3B) and two from biliary tract cancer—HuCC-T1 derived from an intrahepatic biliary tract cholangiocarcinoma (iCCA) and EGI-1 derived from extrahepatic biliary tract CCA (eCCA). All cell lines were maintained at 37 °C in a 5% CO_2_ environment. Details of the media used were reported in the Appendix A. HepG2 cells were purchased from the European Collection of Authenticated Cell Cultures, Hep3B cells were from the American Type Culture Collection, HuCC-T1 cells were from the Japanese Collection of Research Bioresources Cell Bank, and EGI-1 cells were from the DSMZ-German collection of Microorganisms and Cell cultures GmbH. All cell lines were shipped with their authentication certificate. 

### 2.2. Spheroids Formation 

Depending on the experimental assay, cells from all cell lines were seeded with different densities in plates coated with polyhydroxyethylmethacrylate (polyHEMA) 30 mg/mL (Merck Life Science S.r.l., Milan, Italy) and incubated for 3 days at standard conditions (37 °C and 5% CO_2_). To generate spheroids from HepG2 and Hep3B cells, the same growth medium used for two-dimensional (2D) cultures was used, but with FBS 2%. For HuCC-T1 and EGI-1 spheroids, serum-free DMEM/F12 (Gibco, Thermo Fisher Scientific, Waltham, MA, USA) with 1x B27 (Gibco, Thermo Fisher Scientific, Waltham, MA, USA), 10 ng/mL EGF (Merck Life Science S.r.l., Milan, Italy), 10 ng/mL bFGF (Life Technologies, Waltham, MA, USA), and 1% Penicillin-Streptomycin was used. At the end of the formation period, spheroids displayed different sizes among cell lines: Hep3B- and HuCC-T1-derived spheres were smaller than HepG2 and EGI-1 ones (Figure 1A–D). 

### 2.3. The 3D Viability Assay

Two thousand HCC cells/well or 1500 CCA cells/well were seeded into polyHEMA-coated 96-well culture plates. Upon formation, spheroids were stimulated immediately (time 0) with recombinant human ANG-2 (rh-ANG-2) (623-AN, R&D Systems, Minneapolis, MN, USA) and/or recombinant human VEGF (rh-VEGF) (293-VE, R&D Systems) proteins at increasing doses (0, 50, 100, 200, 400, 800 ng/mL). After 48 h from stimulation, the 3D viability assay was performed by transferring the obtained spheroids to a blank 96-well plate without coating and then adding the endpoint CellTiter-Glo^®^ 3D reagent (Promega, Milan, Italy) to induce the ATP release from spheroids, according to the manufacturer’s instructions. The luminescence signal was recorded after 30 minutes with the GloMax^®^ bioluminescent reader (Promega). 

### 2.4. Migration and Invasion Assay

Fifty-thousand cells/well were seeded into a 6-well plate coated with polyHEMA and the obtained spheroids were transferred into 6 or 24 plates without coating to allow attachment and stimulated with 200 ng/mL of rh-ANG-2 or rh-VEGF or a combination of 100 ng/mL of each protein (central doses of the wide range tested in viability assay). Unstimulated spheroids served as the controls. 

In the migration assay, to give the spheroids enough time to adhere, we started the observation 3 h after stimulation and continued it after 24 and 48 h. The end of the observation period was set to 48 h when all spheroids showed a clearly visible migrated area and a reduced necrotic core. In some experiments, inhibitors were added simultaneously by the stimulation with rh-ANG-2 or rh-VEGF. Trebananib (Biogem, Ariano Irpino (AV), Italy) was used to inhibit the effect of rh-ANG-2 (7.5 ng/μL on HuCC-T1 spheroids and 14 ng/μL on HepG2 spheroids), and Bevacizumab (Selleckchem, Houston, TX, USA) was used to inhibit the effect of rh-VEGF (20 ng/mL on Hep3B spheroids). The setting of experimental conditions was obtained by calculating the IC70 for each cell line. Images were taken with a MICA microscope (Leica, Wetzlar, Germany) in widefield mode. We measured the spheroid’s area using the Fiji ImageJ tool as an indicator of migration capability, as our models did not clearly distinguish between the spheroid core and the migration area. After performing statistical analysis, we represented all measurements in a bar graph.

For the invasion assay, spheroids were embedded in a 1:1 mixture of Matrigel 5 mg/mL (Corning, Glendale, AZ, USA) and a cell culture medium supplemented with rh-ANG-2 and/or rh-VEGF proteins in the same amounts as above. Within each replicate, 4–5 spheroids were identified as representative for each condition. We followed individual spheroids over time for quantitative size analysis. We captured images using a Leica DM camera immediately after stimulation (at time 0, serving as the baseline), as well as at 24 and 48 h thereafter. These images were then processed using Fiji ImageJ, and the invasiveness was quantified by measuring the area occupied by spheroids in Matrigel. This approach was necessary as our models did not clearly distinguish between the spheroid core and the invasive area. We log2-transformed the raw data and presented them as fold changes relative to the baseline values.

### 2.5. Gene Expression Analysis

Four hundred thousand cells/well were seeded into a 6-well plate coated with polyHEMA to obtain spheroids for the gene expression analysis of E-cadherin (*CDH1*), N-cadherin (*CDH2*), and Vimentin (*VIM*) markers. Spheroids were transferred into a 6-well plate without coating and stimulated for the migration assay. After 3 and 48 h from stimulation (respectively, after adhesion and after extensive migration of cells), total RNA was extracted from spheroids, quantified, and retro-transcripted. cDNA was used in reactions of droplet digital PCR (ddPCR). Details of the extraction and ddPCR were reported in the Appendix A. Copies/μL of each gene target were normalized on the copies of the actin gene used as reference, and the final data were plotted on the graphs after statistical analysis. 

### 2.6. Western Blot Analysis

Two hundred thousand cells/well were seeded into a 6-well plate coated with polyHEMA to obtain enough spheroids to provide a sufficient number of proteins from lysis. Newformed spheroids were transferred into 6-well plates without polyHEMA and stimulated as above. At 3 and 48 h from the stimulation (as in gene expression analysis), spheroids were collected, lysed, and used in Western blot analysis for the detection of E-cadherin, N-cadherin, and Vimentin expression. We assessed how ANG-2 and the VEGFA activate their specific receptors by stimulating spheroid cultures, as detailed previously, and then we collected samples at 15, 30, and 60 minutes intervals. We then prepared lysates from these samples and analyzed them using Western blotting. This technique allowed us to measure the levels of ANG-2, TIE2, phospho-TIE2 (P-TIE2), VEGFA, VEGFR1, and phospho-VEGFR1 (P-VEGFR1). The specifics about the antibodies we used for this analysis can be found in the Appendix A.

### 2.7. Immunofluorescence Analysis

Fifty thousand cells/well were seeded into a 6-well plate coated with polyHEMA and the obtained spheroids were collected, transferred into 6-well plates with coverslips on the bottom, and stimulated as above. At 3 and 48 h from stimulation, the cells were fixed with 4% formaldehyde (Carlo Erba Reagents, Emmendingen, Germany) and used in the immunofluorescence evaluation of E-cadherin and N-cadherin markers. Details were reported in the Appendix A. Slides were visualized using a Leica MICA microscope in confocal mode. 

### 2.8. Statistical Analysis

Results were expressed as means ± SEM (standard error of the mean). Each experiment was repeated at least three times. Differences between groups were tested for statistical significance with a one or two-way ANOVA followed by appropriate post hoc tests for multiple comparisons using GraphPad software (Prism 9, San Diego, CA, USA). The differences were considered significant for a *p*-value of <0.05.

## 3. Results

### 3.1. rh-ANG-2 and rh-VEGF Stimulation Did Not Affect the Viability of HCC and CCA Spheroids

To test whether rh-ANG-2 and rh-VEGF could influence the number of viable HCC or CCA cells in 3D culture, we performed a viability assay of HCC and CCA spheroids. We did not observe significant differences in luminescence, regardless of different conditions of stimulation (Figure 2A–D), suggesting that rh-ANG-2 and rh-VEGF do not affect cell viability. 

### 3.2. rh-ANG-2 and rh-VEGF Stimulation Increased Migration Capability in HepG2, Hep3B, and HuCC-T1 but Not EGI-1 Spheroids

Starting from the well-documented role of ANG-2 and the VEGF in facilitating cellular migration [31,32], we assessed their impact on HCC and CCA spheroids. Our evaluation compared the migration of treated spheroids to that of the control group (untreated) and also examined the changes in migration across various time points post-treatment.

In HepG2 spheroids, bubble formation was noted along the spheroid perimeters as early as 3 h post-stimulation. By the 24 h mark, all spheroids exhibited similar growth. However, after 48 h, the area occupied by migratory cells was significantly larger in spheroids treated with rh-ANG-2 and rh-VEGF compared to the 3 h measurement. Notably, only the rh-ANG-2 treatment led to a significant increase in migration compared to the untreated controls, as shown in Figure 3A,B.

Three hours post-stimulation, the Hep3B spheroids exhibited increased adherence to the plate compared to those derived from HepG2. These treated spheroids displayed a partially visible migration zone around their perimeter. At 24 h, and more notably at 48 h post-stimulation, this migration zone had significantly enlarged in comparison to the 3 h measurement in spheroids treated with rh-VEGF or a combination of rh-ANG-2 and rh-VEGF. The increase in migration was substantially higher in these two groups than the untreated controls (as illustrated in Figure 3C,D). The control spheroids retained their original morphology, and the migration zone remained indistinct.

In spheroids derived from HuCC-T1, a migration zone was evident 24 h post-stimulation, and complete cell migration was observed after 48 h. Notably, the migration zone in all spheroids expanded significantly over time under all tested conditions, with the exception of those treated with both rh-ANG-2 and rh-VEGF. However, only the spheroids treated with rh-ANG-2 alone exhibited a migration zone significantly larger than the control after 48 h (as shown in Figure 3E,F).

EGI-1 spheroids differed from those derived from HuCC-T1; they were slightly larger and more well-defined in shape. Initially, at 3 h, only a few spheroids adhered, and no sprouting cells were detected under any of the conditions. After 24 h, and more pronouncedly at 48 h, the cells dispersed, forming a distinct migration zone. The area of the spheroids increased significantly over time in all conditions (refer to Figure 3G,H). Yet, there were no discernible differences between the control and treated spheroids.

### 3.3. Trebananib and Bevacizumab Effectively Inhibited the rh-ANG-2 and rh-VEGF-Stimulated Migration of Spheroids Derived from HepG2, HuCC-T1, and Hep3B Cell Lines

To further confirm the role of proangiogenic factors in promoting spheroid migration, we inhibited rh-ANG-2 and rh-VEGF stimuli using specific inhibitors, Trebananib and Bevacizumab, respectively. We conducted a migration assay, as previously described, stimulating HepG2 and HuCC-T1 spheroids with rh-ANG-2 and Hep3B spheroids with rh-VEGF in the presence or absence of these specific inhibitors. Observations were made at 3, 24, and 48 h post-stimulation (Figure 4A–F). Concurrently, levels of ANG-2 and VEGFA proteins were quantified in lysates to verify inhibition within the cells (Figure 4G–I). At 24 h for HuCC-T1 (Figure 4C,D) and 48 h for both HepG2 (Figure 4A,B) and HuCC-T1 (Figure 4C,D), Trebananib significantly reduced the migration enhancement caused by rh-ANG-2, with areas of migration substantially smaller than those stimulated with rh-ANG-2 alone. Western blot analysis showed a marked decrease in ANG-2 protein in the rh-ANG-2 and Trebananib-treated spheroids as early as 3 h in HepG2 (Figure 4G) and in HuCC-T1 (Figure 4H), as well as at 24 and 48 h in HepG2 (Figure 4G). 

In Hep3B spheroids, Bevacizumab effectively reduced the rh-VEGF-dependent migration as early as 24 h post-stimulation, with further decrease observed at 48 h (Figure 4E,F). Additionally, VEGFA protein levels in Hep3B spheroids treated with a combination of rh-VEGF and Bevacizumab were significantly reduced at 3 h and remained lower at 24 h post-treatment (Figure 4I).

### 3.4. rh-ANG-2 and rh-VEGF Were Found to Enhance the Invasion Capabilities of HuCC-T1 and Hep3B Spheroids

Following the outcomes of the migration assay, we conducted an invasion assay in Matrigel to determine the impact of proangiogenic stimuli on invasiveness. For HepG2 spheroids, a significant increase in the invasive area was observed at 24 h post-stimulation with rh-ANG-2 or rh-VEGF compared to the baseline. However, at 48 h, only spheroids treated with rh-ANG-2 continued to show a significant expansion (Figure 5A,B). In Hep3B spheroids, the invasive area increased by 24 h post-stimulation with either rh-VEGF alone or in conjunction with rh-ANG-2. Yet, at the 48 h mark, significant enlargement was only evident in spheroids subjected to the individual treatments of rh-ANG-2 or rh-VEGF (Figure 5C,D). 

In HuCC-T1 spheroids, a significant enlargement in area was observed 24 h after stimulation with rh-ANG-2. By the 48 h mark, spheroids exposed to rh-ANG-2 alone and in combination with rh-VEGF exhibited areas significantly larger than the baseline (Figure 5E,F). In EGI-1, both the control and stimulated spheroids displayed a significant area increase at 24 h, which persisted at 48 h, regardless of the stimulation condition (Figure 5G,H).

Across the four cell lines examined, only Hep3B and HuCC-T1 demonstrated a notable effect on invasive capability when comparing treated spheroids with the controls, each responding to different proteins. Specifically, rh-VEGF enhanced the invasiveness in Hep3B spheroids at 48 h post-stimulation (Figure 5C,D), whereas in HuCC-T1, this effect was attributed to rh-ANG-2 (Figure 5E,F). These findings are in line with the migration assays, corroborating the role of rh-ANG-2 in HuCC-T1 and rh-VEGF in Hep3B as facilitators of cellular migration and invasion.

### 3.5. Assessment of ANG-2 and VEGFA Receptor Expression and Activation

Our migration assays, conducted in the presence and absence of inhibitors, highlighted the significant role of rh-ANG-2 and rh-VEGF in enhancing the migratory capacity of HepG2 and HuCC-T1 spheroids, as well as Hep3B spheroids, respectively. Subsequently, we examined the expression and activation of TIE2 and VEGFR1, their receptors, and their phosphorylated counterparts, P-TIE2 and P-VEGFR1. These were detected in HepG2, HuCC-T1, and Hep3B spheroids at time intervals of 15, 30, and 60 min following stimulation, as shown in Figure 6. For HepG2, TIE2 expression was consistent across treatments at 15 min. We noted a modest increase at 30 and 60 min, particularly in spheroids treated with rh-ANG-2 compared to the control group at the 30 min mark. The activation of TIE2, as indicated by its phosphorylation, was significantly higher in the rh-ANG-2-treated spheroids starting from 15 min. This increase in activation was confirmed by comparing the ratio of phosphorylated TIE2 to total TIE2 (Figure 6A,B). In HuCC-T1 spheroids, TIE2 expression remained unchanged at the earlier time points of 15 and 30 min but showed a significant increase at 60 min when stimulated with rh-ANG-2. Notably, despite the stable TIE2 levels, phosphorylation levels (indicating receptor activation) did not show marked changes at 15 and 30 min. However, they were significantly reduced at 60 min following stimulation with rh-VEGF alone and in combination with rh-ANG-2 compared to the controls. This reduction was even more pronounced when both proangiogenic factors were used together, which was also reflected in the P-TIE2 to total TIE2 ratio. Interestingly, there was a significant decrease in P-TIE2 levels at 60 min with rh-ANG2 stimulation, which was attributed to increased receptor expression without a corresponding increase in phosphorylation status (as shown in Figure 6C,D). For Hep3B spheroids, VEGFR1 expression was consistently maintained across different conditions and over time. The activation of VEGFR1, as determined by phosphorylation, was apparent at 60 min in spheroids treated with rh-VEGF compared to the untreated controls. This finding was further supported by the significant ratio of P-VEGFR1 to total VEGFR1, as depicted in Figure 6E,F. It is noteworthy that the Western blot analysis of VEGFR2 expression in Hep3B spheroids resulted in ambiguous outcomes, characterized by non-specific signals that were unquantifiable, and thus these data are not presented. Analysis of TIE2 and VEGF ligands (ANG-2 and VEGFA, respectively) is reported in the Appendix A.

### 3.6. Differential Effects of rh-ANG-2 and rh-VEGF on EMT in HCC and CCA Cell Lines

To ascertain the impact of rh-ANG-2 and/or rh-VEGF stimulation on EMT marker expression, we conducted both gene and protein analyses for E-cadherin (*CDH1*), N-cadherin (*CDH2*), and Vimentin (*VIM*) in lysates derived from spheroids. Transcriptomic analysis is reported in the Appendix A.

At the protein level, the HepG2 cell line demonstrated a notable decrease in E-cadherin expression as early as 3 h post-treatment across all conditions compared to the controls, accompanied by an increase in N-cadherin expression, with the exception of rh-ANG-2 stimulated spheroids. These alterations in protein expression remained consistent at 48 h, showing no significant fluctuations among the treated groups. Vimentin levels were significantly elevated in spheroids treated with rh-VEGF and the combination of rh-ANG-2 and rh-VEGF at 48 h compared to the controls and to the same treatments at 3 h. For Hep3B spheroids, a significant reduction in E-cadherin was observed at 48 h in spheroids treated with both stimuli compared to the 3 h mark, which corresponded to increases in both N-cadherin and Vimentin. These last markers also showed a significant rise with rh-VEGF treatment alone. In HuCC-T1 cells, only Vimentin showed a significant increase at 48 h when stimulated with rh-ANG-2, relative to the 3 h measurement. Conversely, in EGI-1 spheroids, E-cadherin levels remained constant regardless of treatment or time point, and both N-cadherin and Vimentin were not detected, which may be attributable to the low gene expression levels noted (Figure 7 and Appendix A).

### 3.7. The E-Cadherin to N-Cadherin Switch, Indicative of EMT Activation, Was Observed in the Outermost and Migrating Cells of the Spheroids

We conducted an immunofluorescence assay to discern the localization and signal intensities of E-cadherin and N-cadherin within spheroids post-stimulation. This technique was selected over Western blotting, which does not differentiate between non-migrated and proliferating cells migrating within the 3D spheroid structure, as referenced in [33]. Due to the absence of observable differences among the three treatments in this assay, we have consolidated the results, treating the ‘treated’ condition as a collective representation of all three treatment modalities. At 3 h post-treatment, untreated HepG2 spheroids exhibited a uniform expression of E-cadherin, while peripheral cells of the treated spheroids showed diminished expression. By 48 h, this reduction in E-cadherin expression was more pronounced in the migrating cells of both treated and untreated spheroids, as highlighted by red arrows in Figure 8A. Similarly, Hep3B spheroids demonstrated a decline in E-cadherin expression as early as 3 h post-treatment and more noticeably at 48 h in stimulated spheroids compared to the controls, particularly in the outer and migrating cells (Figure 8B).

N-cadherin expression was low in both HCC cell lines at 3 h post-stimulation and was not influenced by treatment. However, at 48 h, we observed a widespread increase in intensity, with treated spheroids showing stronger N-cadherin expression than the controls (Figure 8A for HepG2 and Figure 8B for Hep3B).

In HuCC-T1 spheroids, there was a marked reduction in E-cadherin expression at 48 h, regardless of treatment, especially at the migrating front (red arrows in Figure 8C). 

Concurrently, a minor elevation in N-cadherin levels was noted. For EGI-1 spheroids, consistent E-cadherin expression was observed at both 3 and 48 h post-stimulation, with no discernible difference between stimulated and unstimulated cells. Notably, at 48 h, the E-cadherin signal varied, showing higher intensity within regions where cells maintained a compact structure and lower intensity among migrating cells (Figure 8D). EGI-1 spheroids did not exhibit N-cadherin expression under any experimental conditions, which was consistent with Western blot results.

## 4. Discussion

In studies on HCC and to a somewhat lesser extent, iCCA, the contributions of ANG-2 and the VEGF to the advancement of these cancers have been thoroughly documented through analysis of patient samples, including serum and tissue biopsies. High levels of these proteins, along with increased activity of their genes, highlight their critical roles in angiogenesis and neo-angiogenesis. These processes are crucial for the growth of cancer and contribute to making the disease more severe. However, there is a lack of extensive research on how ANG-2 stimulation affects HCC and CCA, whether it is applied by itself or along with the VEGF. Previous studies have shown that the VEGF directly affects HCC [31,34] and CCA [35,36] in laboratory settings. While the individual impacts of the VEGF have been documented, the combined effects with ANG-2, especially known to work together to influence blood vessel cells in cancer, have not been fully explored and deserve more attention [37,38].

Recent research by Rawal et al. [39] has highlighted that liver cancer cells behave differently when grown in 3D cultures compared to traditional flat (2D) cultures, indicating that 3D models might better replicate the actual conditions of cancer growth and its surrounding environment. This idea is gaining support from more studies in the field [33,40]. Based on these insights, we have developed a 3D culture system using cell lines from HCC—HepG2 and Hep3B—as well as CCA cell lines—HuCC-T1 and EGI-1. Our aim was to thoroughly examine how these cells react to factors that promote blood vessel formation, which is a key process in cancer development.

For HCC models, we utilized spheroids from the HepG2 and Hep3B cell lines to represent less and more aggressive tumor types, respectively. Comparative analyses of gene expression, drug responsiveness, and signaling pathways indicate that Hep3B exhibits a markedly more aggressive phenotype than HepG2 [41]. In the case of CCA, spheroids were derived from HuCC-T1 and EGI-1 cell lines to represent intrahepatic and extrahepatic tumors, respectively. Studies of global gene expression suggest that the HuCC-T1 cell line is similar to HepG2 in many ways [42]. Meanwhile, other research has noted that HuCC-T1 has comparable characteristics to the Hep3B cell line, especially in how they both respond to cancer drugs that inhibit cell growth [43]. Based on these findings, we decided that the HuCC-T1 cell line was a good choice for testing our ideas. For the study of extrahepatic CCA, we chose the EGI-1 cell line, which has been extensively studied and characterized in the literature [44,45,46], despite the limited availability of CCA-derived cell lines.

Firstly, we assessed whether stimulation with rh-ANG-2 and/or rh-VEGF affected cell viability, irrespective of the concentration used, to ensure that the stimulation did not exert toxic side effects. We utilized an endpoint assay to evaluate only the number of viable cells producing ATP. It is important to note that assessing viability does not equate to assessing proliferative capacity, which can be altered by the spheroid structure; indeed, cells at the center of a spheroid remain viable but not proliferative [47]. 

The migration assay revealed that spheroids from all cell lines exhibited increased motility in response to rh-ANG-2 and rh-VEGF, albeit to varying degrees. Specifically, the HepG2 and HuCC-T1 cell lines displayed comparable responses to rh-ANG-2 stimulation. Conversely, Hep3B spheroids showed no significant response to rh-ANG-2 but had an increased response to rh-VEGF. Although the synergistic effects of ANG-2 and the VEGF are well-documented [19,48], we found that combining rh-ANG-2 with rh-VEGF did not increase migration in either HepG2 or iCCA spheroids. This observation implies that ANG-2 may counteract the VEGF’s effects, which is consistent with other cancer models where ANG-2 stimulation led to reduced migratory capabilities normally induced by the VEGF [49,50]. Similarly, in Hep3B cells, migration that was dependent on rh-VEGF did not show further enhancement with the addition of rh-ANG-2 suggesting that rh-VEGF is the primary driver of migration in this cell line. This pro-migratory effect aligns with the findings reported by Sharma et al. [51], in which the authors demonstrated a decrease in Hep3B migration following VEGF silencing.

The hypothesis that ANG-2 or the VEGF could stimulate migration gained support when introducing specific inhibitors—Trebananib for ANG-2 and Bevacizumab for the VEGF. These inhibitors not only suppressed the pro-migratory effect but also decreased the expression levels of the respective proteins, as verified by Western blot analysis. Contrary to expectations, EL-Hajjar and colleagues demonstrated that Bevacizumab treatment increased migration and invasion in aggressive cancer cell lines and also elevated pro-inflammatory markers [52]. This Bevacizumab-associated pro-inflammatory effect has been similarly noted at the intraocular level [53]. However, other studies have confirmed Bevacizumab’s efficacy in reducing inflammation [54,55]. Given the significant link between liver cancer, especially HCC, and inflammation [56,57], further investigation into the inflammatory mediators after Bevacizumab treatment is justified. Notably, EGI-1 spheroids consistently demonstrated increased motility, irrespective of the stimulation. It is worth noting that the control spheroids exhibited migration and growth in all cell lines, with the most pronounced effects observed in EGI-1. This evidence suggests that the migratory mechanisms may be inherently linked to the spheroid culture conditions, especially influenced by the high migration rate of the cells at the periphery [58]. The invasion assay corroborated the differential impacts of rh-ANG-2 and rh-VEGF on HuCC-T1 and Hep3B spheroids, respectively, as previously observed in the migration assays. While only HuCC-T1 showed a significant difference in response to rh-ANG-2 compared to the control, a general trend of increased invasive capacity over time was observed across all cell lines, except for EGI-1, which consistently exhibited higher invasiveness, regardless of time and stimulation. The capacity of the VEGF to promote proliferation and invasiveness in both endothelial and non-endothelial cells is well-documented [59]. For instance, in ovarian cancer, the VEGF has been shown to expedite tumor cell invasion into Matrigel by facilitating the degradation of the extracellular matrix [60]. Similarly, in T-47D breast cancer cells, the VEGF markedly enhanced cellular invasion through Matrigel and fibronectin-coated transwell membranes [61]. The role of ANG-2 in this context remains less understood; nevertheless, several studies have suggested a correlation between escalated doses of ANG-2 and increased proliferation in several estrogen receptor-positive (ER+) breast cancer cell lines [62]. Similarly, in glioma and extravillous trophoblast cell lines, the addition of exogenous ANG-2 has been associated with heightened invasiveness [21,63]. Similar to the migration assay, combining rh-ANG-2 and rh-VEGF did not produce a discernible effect in the invasion assays for either Hep3B or HuCC-T1 cells. This lack of effect implies that these factors might act as antagonists in our tumor models.

To investigate potential mechanisms underlying our observed effects on migration and invasiveness, we examined the expression and phosphorylation of canonical receptors for ANG-2 and the VEGF following proangiogenic stimulation. We analyzed the expression of TIE2 and its phosphorylated form (P-TIE2) in HepG2 and HuCC-T1 spheroids and VEGFR1 and its phosphorylated form (P-VEGFR1) in Hep3B spheroids. The significant increase in the P-TIE2/TIE-2 ratio, indicative of enhanced phosphorylation, suggests a direct action of rh-ANG-2 on TIE2 in HepG2, contributing to its pro-migratory behavior. In various conditions, such as HCC [64], ANG-2’s activation of the TIE2 receptor has been associated with HCC neovascularization and disease progression pathways [65]. Therefore, the observed increase in migration in HepG2 spheroids treated with rh-ANG-2 is likely mediated by the ANG-2/TIE-2 axis. However, in HuCC-T1, despite a significant increase in TIE2 expression, the lack of enhanced phosphorylation by rh-ANG-2 suggests that this axis may not be directly involved in acquiring pro-migratory/pro-invasive capabilities. The decreased P-TIE2/TIE2 ratio after rh-ANG-2 treatment could indicate that TIE2 was not activated. In tumoral cells, ANG-2 has been shown to bind and signal through other types of receptors, such as integrins, resulting in glioma cell invasion [66,67] and increased breast cancer cell survival and invasion [68]. Exploring this pathway in our models could be a promising direction for future investigations. The reduction in TIE2 phosphorylation observed in spheroids stimulated with rh-VEGF alone and in combination with rh-ANG-2 may suggest a potential inhibitory effect of rh-VEGF on TIE2 activation. This finding aligns with the results of a study by Findley and colleagues [69], who demonstrated that the VEGF induces the shedding of TIE2, leading to a subsequent loss of function in endothelial cells.

In Hep3B, we investigated the expression and phosphorylation-based activation of VEGFR1, as this cell line exhibited a prominent pro-migratory and pro-invasive response to rh-VEGF treatment. Despite not being the best-characterized VEGFA receptor, VEGFR1 is known to be expressed in various cancer cells and is associated with increased invasive capabilities in numerous studies [22,23,70,71]. HCC cell lines also express VEGFR1, which appears to be linked to cell motility and invasion [72,73]. Moreover, its presence in human HCC tissue is correlated with tumor progression [72]. Our findings confirm that rh-VEGF activates VEGFR1 in our 3D model of Hep3B, suggesting that rh-VEGF binds to and activates this receptor on the cells, thereby enhancing invasive capabilities.

As EMT is a well-recognized mechanism that facilitates cell migration [74], we conducted an investigation on the impact of proangiogenic stimuli, specifically ANG-2 and the VEGF, on the EMT phenotype using Western blot analysis. In HepG2 spheroids, significant changes in EMT markers were observed as early as 3 h and as late as 48 h after treatment, marked by a notable decrease in E-cadherin and an increase in N-cadherin and Vimentin. Conversely, Hep3B spheroids exhibited changes in these markers, primarily at later stages, particularly in response to rh-VEGF, and at the protein level. In the case of HuCC-T1 spheroids, a consistent increase in Vimentin expression was observed; however, this change was associated with rh-ANG-2 treatment at the protein level, despite its presence under all conditions at the gene level. EGI-1 spheroids exhibited a unique profile, showing increased migration and invasiveness, along with a spontaneous elevation in EMT marker expression. This was characterized by stable E-cadherin levels and a lack of N-cadherin or Vimentin expression. These findings were corroborated by immunofluorescent analysis, which highlighted the EMT changes induced by proangiogenic stimuli, particularly in HepG2 and Hep3B spheroids. Collectively, our data underscore the propensity of cell line-derived spheroids to undergo EMT, albeit with varying responses to ANG-2, the VEGF, or their combination. Notably, HuCC-T1 cells exhibited a more gradual reaction to stimuli, with Vimentin being the sole marker showing significant alteration. This observation suggests the possibility of a ‘partial’ EMT phenomenon, akin to what has been described in other cancer models, such as squamous carcinoma and breast cancer [75,76], which has been associated with increased metastatic capacity [77,78]. Furthermore, Vimentin’s role in promoting metastasis was supported by its identification as a critical factor in the metastatic potential of aggressive CCA cell lines in vitro [79]. This aligns with the established correlation between aberrant Vimentin expression in CCA tissues and the dedifferentiation and poor prognosis in patients [80,81]. The response of HuCC-T1 spheroids to proangiogenic stimuli corroborates these findings. In our examination of the spatial distribution of E-cadherin and N-cadherin using immunofluorescence, we discerned a uniform pattern among all cancer cell models studied. Specifically, there was a noticeable decrease in E-cadherin and an increase in N-cadherin in the outer migrating layer of the spheroids. This observation aligns with the established concept that EMT primarily occurs at the invasive front of tumors, where cells tend to lose their cell-cell contacts [33,81,82]. Intriguingly, even the untreated spheroids gradually exhibited changes in EMT marker expression over time, lending support to the notion that the spheroidal culture conditions might inherently trigger this mechanism. It seems that a Western blot analysis of the whole spheroid lysate may not accurately capture the EMT marker changes occurring at the spheroid’s periphery. Although immunofluorescence analysis enhanced our understanding of the variations in marker expression, particularly at the spheroid’s edge, it is a qualitative method. Consequently, it lacked the sensitivity to distinguish nuanced differences attributed to the various stimulations.

In conclusion, our data indicate that spheroids derived from two primary HCC cell lines and one iCCA cell line exhibit both similarities and notable differences in response to proangiogenic stimuli. While a direct role of rh-ANG-2 and rh-VEGF can be assumed, the observed variations highlight the complexity of their effects in these distinct cell line models. Future directions will include the evaluation of the reactivity to proangiogenic stimuli by primary cell lines derived from human HCC and iCCA as a prerequisite to testing them in patient-derived xenografts.

## Figures and Tables

**Figure 1 biomedicines-12-00087-f001:**
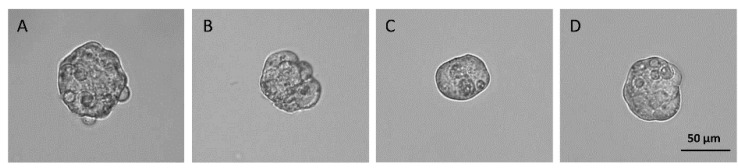
Spheroids of HCC and CCA cell lines. Examples of spheroids derived from HepG2 (**A**), Hep3B (**B**), HuCC-T1 (**C**), and EGI-1 (**D**) after formation. The images were acquired at the magnification of 20× at the same time point. The same number of cells were seeded for all cell lines. Depending on the aggregation mechanism of the cells, the size of spheroids appeared slightly different. HepG2 and EGI-1 spheroids were slightly larger, while Hep3B and HuCC-T1 spheroids were smaller.

**Figure 2 biomedicines-12-00087-f002:**
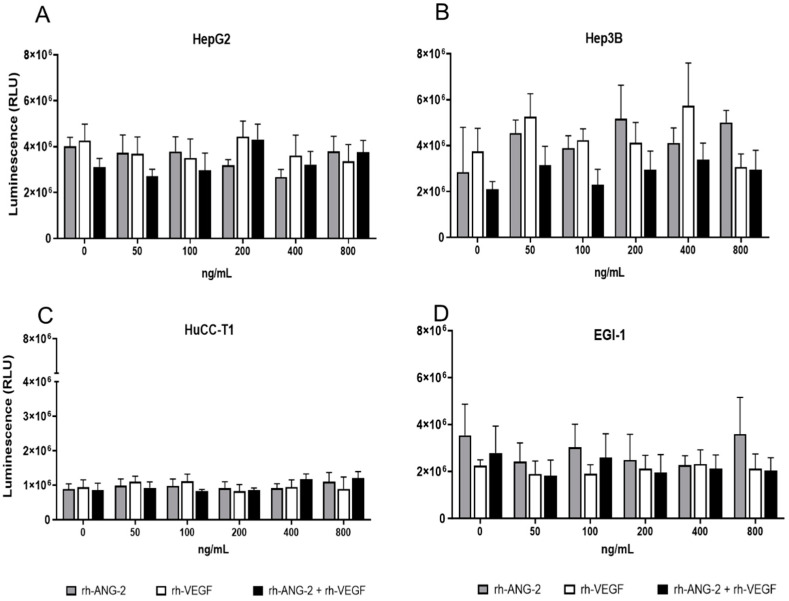
The 3D viability assay. This assay was conducted for HepG2 (**A**), Hep3B (**B**), HuCC-T1 (**C**), and EGI-1 (**D**), which were stimulated with concentrations of 0, 50, 100, 200, 400, and 800 ng/mL of rh-ANG-2 and rh-VEGF, either alone or in combination, for a duration of 48 h. The viability within the 3D cell culture was assessed by quantifying ATP and is indicative of metabolically active cells. ATP levels were represented in histograms as luminescence signals (RLUs). All cell lines exhibited comparable luminescence levels, with the exception of HuCC-T1, which demonstrated lower levels (HepG2: 3.6 × 10^6^; Hep3B: 3.8 × 10^6^; HuCC-T1: 9.6 × 10^5^; EGI-1: 2.4 × 10^6^ luminescence units). The experiments were performed in triplicate. Statistical significance between groups was determined using a one-way ANOVA. For all comparisons within different groups, no significant differences were present between untreated (0 ng/mL) and treated spheroids (*p* > 0.05).

**Figure 3 biomedicines-12-00087-f003:**
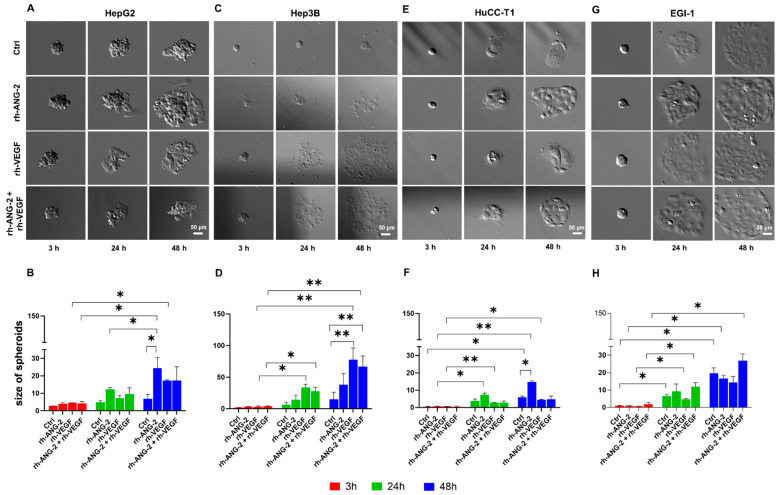
Migration assay of HCC and CCA spheroids. Migration assays were performed on spheroids from HepG2 (**A**,**B**), Hep3B (**C**,**D**), HuCC-T1 (**E**,**F**), and EGI-1 (**G**,**H**) cell lines. These were either untreated (Ctrl) or treated with 200 ng/mL of rh-ANG-2, 200 ng/mL of rh-VEGF, or a combination of 100 ng/mL each of rh-ANG-2 and rh-VEGF. Measurements were taken at 3, 24, and 48 h post-treatment. Due to size reduction and cropping, images of EGI-1 spheroids are larger than the frame. Spheroid areas were quantified using Fiji ImageJ to assess migration capability (HepG2 (**B**), Hep3B (**D**), HuCC-T1 (**F**), EGI-1 (**H**)). Imaging was performed at 10× magnification. Comparative analyses were conducted to assess differences between the control and treated spheroids at the same time points. We also compared different areas of the spheroids at 24 and 48 h, in contrast to the changes observed at 3 h within the same treatment group. The assays were replicated a minimum of three times. Statistical differences were evaluated using a two-way ANOVA for paired data, with subsequent post hoc testing for multiple comparisons (* *p* < 0.05, ** *p* < 0.01).

**Figure 4 biomedicines-12-00087-f004:**
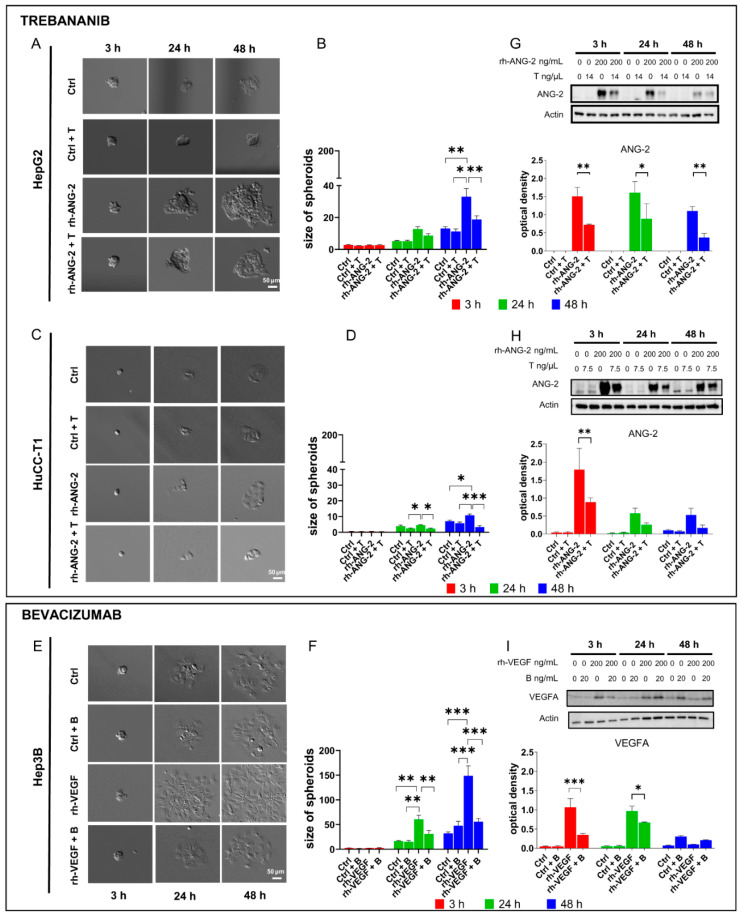
Migration assay and Western blot of ANG-2 and VEGFA expression following treatment with Trebananib and Bevacizumab. HepG2 (**A**) and HuCC-T1 (**C**) spheroids were left without proangiogenic stimulation (Ctrl and Ctrl plus Trebananib (T)) or treated with 200 ng/mL of rh-ANG-2 either alone or in combination with Trebananib (T). Measurements were taken at intervals of 3, 24, and 48 h post-treatment. The spheroid area was quantified using Fiji ImageJ, serving as a proxy for migration capability (HepG2 (**B**), HuCC-T1 (**D**)). For Hep3B spheroids (**E**), which were without proangiogenic stimulation (Ctrl and Ctrl plus Bevacizumab (**B**)) or treated with 200 ng/mL of rh-VEGF alone or with Bevacizumab (**B**); measurements were taken at the same time intervals. The spheroid area, indicative of migration capability, was quantified using Fiji ImageJ (Hep3B (**F**)). We made comparisons between the spheroids that were treated with rh-ANG-2 or rh-VEGF alone and other treatment groups, evaluating them at the same time intervals. All images were captured at a 10× magnification. Due to the reduction and cropping of images, Hep3B spheroids treated solely with rh-VEGF at 48 h surpass the image boundaries. Protein expression was analyzed in HepG2 (**G**) and HuCC-T1 (**H**) spheroids treated with rh-ANG-2 alone or with Trebananib (T) and in Hep3B (**I**) spheroids treated with rh-VEGF alone or with Bevacizumab (**B**) at 3, 24, and 48 h post-stimulation. We quantified the protein bands using ImageLab 6.1 software. The results from the densitometry are reported as optical density, which we have normalized to β-actin to ensure accurate comparisons. A comparison was made between groups treated with rh-ANG-2 or rh-VEGF alone versus those treated with the addition of Trebananib or Bevacizumab at each time point. These experiments were replicated at least three times to ensure reliability. Statistical differences were determined using a two-way ANOVA, followed by post hoc tests for multiple comparisons. Significance levels are denoted by asterisks: * *p* < 0.05; ** *p* < 0.01; *** *p* < 0.001.

**Figure 5 biomedicines-12-00087-f005:**
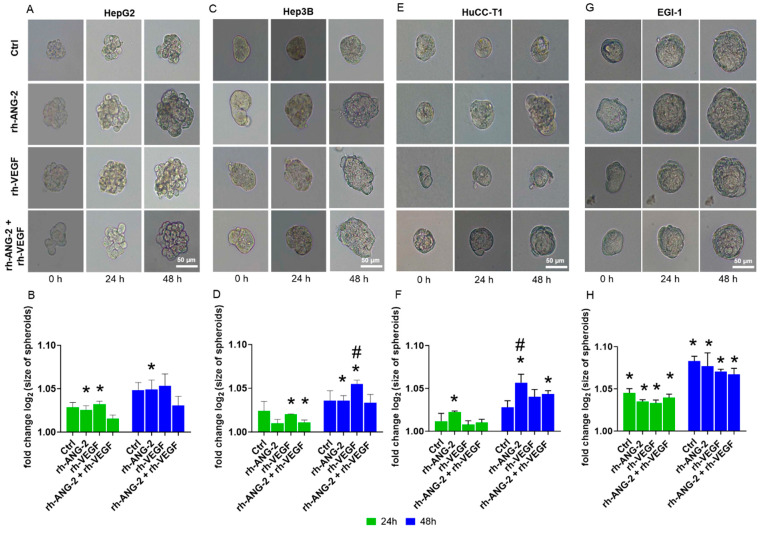
Invasion assay of HCC and CCA spheroids in Matrigel. Spheroids from HepG2 (**A**,**B**), Hep3B (**C**,**D**), HuCC-T1 (**E**,**F**), and EGI-1 (**G**,**H**) were treated with 200 ng/mL of rh-ANG-2 or rh-VEGF independently and with a combination of 100 ng/mL of rh-ANG-2 and rh-VEGF. Imaging was conducted at 0, 24, and 48 h post-stimulation to monitor the invasion process. The degree of invasiveness of the spheroids was quantified by Fiji ImageJ software. We measured the area they covered after being embedded in Matrigel. Comparative analyses were carried out between the control (Ctrl) and treated spheroids at the same time points. Additionally, within each treatment group, we compared the area of the spheroids at 24 and 48 h with their initial size at time 0. These experiments were performed in triplicate to ensure the robustness of the data. The results are presented as the ratio of the treated spheroids’ area to the baseline, with log2 transformation applied to raw data for normalization. The baseline value was standardized at Y = 1.0. Statistical differences were assessed using a two-way ANOVA for paired data, complemented by post hoc tests for multiple comparisons. Significance is indicated by asterisks (*) for *p* < 0.05 compared to the baseline within each treatment group and hashes (#) for *p* < 0.05 compared to the control at the same time points.

**Figure 6 biomedicines-12-00087-f006:**
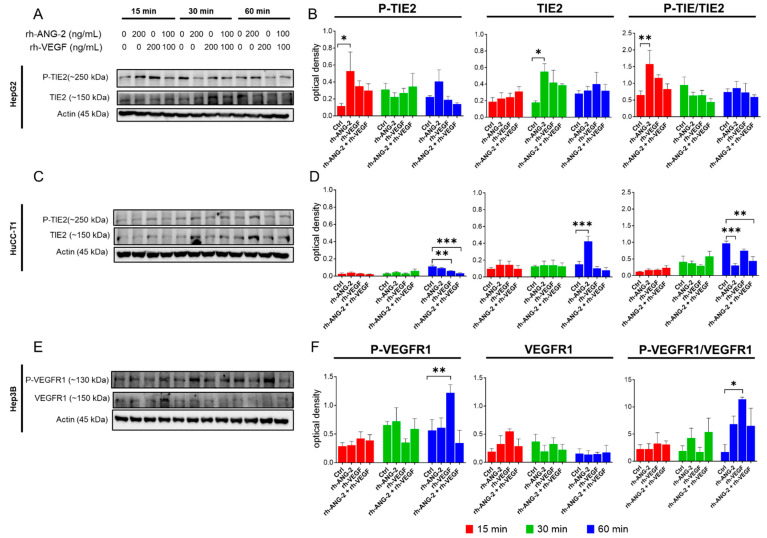
Western blot for TIE2 and VEGFR1 receptor expression and activation by phosphorylation. We evaluated the expression of TIE2 and P-TIE2 in HepG2 and HuCC-T1 spheroids (**A**–**D**) and the expression of VEGFR1 and P-VEGFR1 in Hep3B spheroids (**E**,**F**). Treatments included 200 ng/mL of rh-ANG-2, 200 ng/mL of rh-VEGF, and a combination of 100 ng/mL of rh-ANG-2 and rh-VEGF. Proteins were harvested at 15, 30, and 60 min post-stimulation. Band intensities were quantified using ImageLab 6.1 software, and densitometry data were presented as optical density values normalized to β-actin. Comparative analyses were performed between the treated spheroids and the controls at identical time points. This experiment was replicated a minimum of three times. Statistical differences were assessed using a two-way ANOVA, with subsequent post hoc tests for multiple comparisons. Significance thresholds were established as follows: * *p* < 0.05; ** *p* < 0.001; *** *p* < 0.001.

**Figure 7 biomedicines-12-00087-f007:**
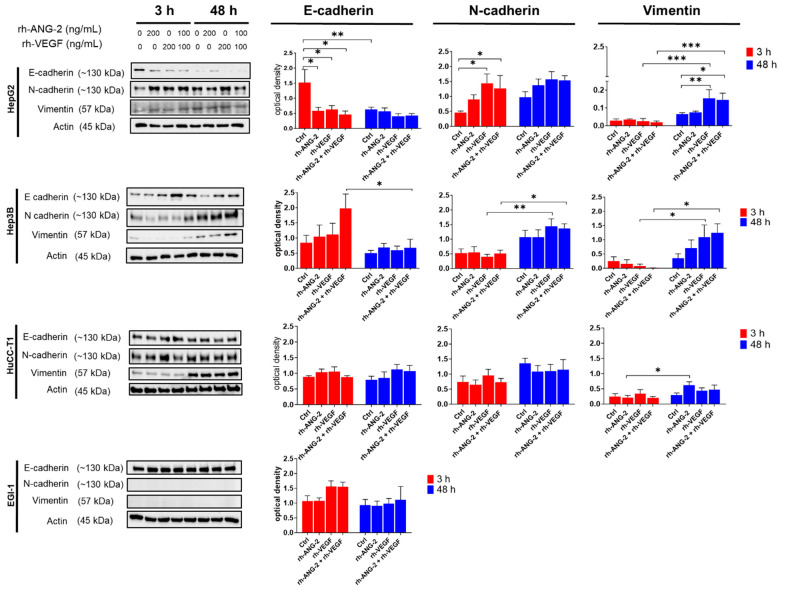
Western blot for EMT marker expressions in HCC and CCA cell lines. We evaluated the expression of E-cadherin, N-cadherin, and Vimentin in HepG2, Hep3B, HuCC-T1, and EGI-1 spheroids. Treatments included 200 ng/mL of rh-ANG-2, 200 ng/mL of rh-VEGF, and a combination of 100 ng/mL of rh-ANG-2 and rh-VEGF. Proteins were harvested at 3 and 48 h post-stimulation. Band intensities were quantified using ImageLab 6.1 software, and densitometry data were presented as optical density values normalized to β-actin. Comparative analyses were performed between the treated spheroids and the controls at identical time points, as well as longitudinally within treatment groups from 3 to 48 h. This experiment was replicated a minimum of three times to ensure reproducibility. Statistical differences were assessed using a two-way ANOVA, with subsequent post hoc tests for multiple comparisons. Significance thresholds were established as * *p* < 0.05; ** *p* < 0.001; *** *p* < 0.001. Comprehensive EMT marker data are summarized in Appendix A.

**Figure 8 biomedicines-12-00087-f008:**
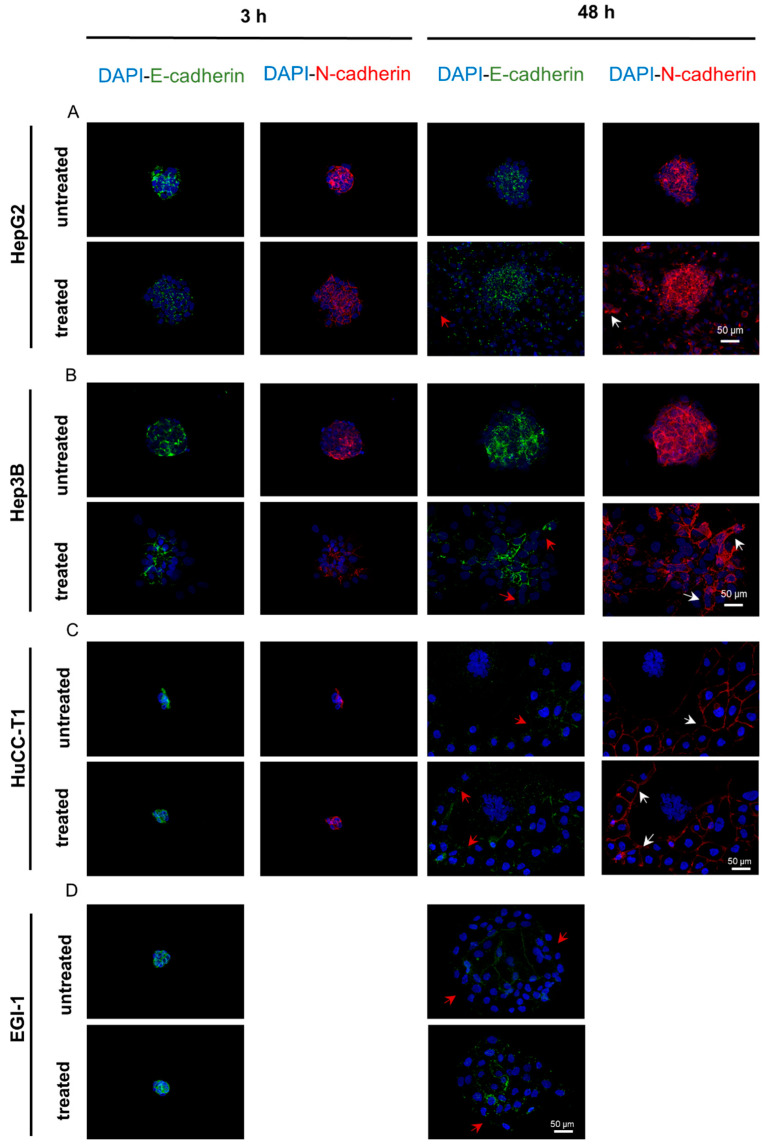
Immunofluorescence of EMT markers in HCC and CCA spheroids. Representative images of HCC and CCA spheroids, either untreated or subjected to proangiogenic stimuli, at 3 and 48 h post-stimulation. We examined spheroids derived from (**A**) HepG2, (**B**) Hep3B, (**C**) HuCC-T1, and (**D**) EGI-1 cell lines. E-cadherin was visualized using a green fluorescent dye, while N-cadherin was detected with a red fluorescent dye. Cell nuclei were counterstained with DAPI. The loss of E-cadherin and the upregulation of N-cadherin expression are denoted by red and white arrows, respectively.

## Data Availability

All data supporting the findings of this study are available within the paper and its Appendix A.

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
