# Peer review of "Angiopoietin-2 and the Vascular Endothelial Growth Factor Promote Migration and Invasion in Hepatocellular Carcinoma- and Intrahepatic Cholangiocarcinoma-Derived Spheroids"

_biomedicines, 2023, doi:10.3390/biomedicines12010087_

Round 1

Reviewer 1 Report (Previous Reviewer 1)

Comments and Suggestions for Authors

In the manuscript titled “ANG-2 and VEGF promote migration and invasion in hepatocellular carcinoma- and intrahepatic cholangiocarcinoma-derived spheroids”, Adriana et al investigated the regulation of HCC and iCCA-derived spheroids to migrate and invade through ANG-2 and VEGF. This manuscript is well written, with only a few comments.

1.     In Figure 2, authors should provide probability of stastical analysis with p-value.

2.     In Figure 6, there are differences between the bands and the graph. For example,
- In (A) 15 min, P-TIE2, “rh-VEGF” is more expressed than “rh-ANG-2”. However, in (B), “rh-ANG-2” shown higher expression than “rh-VEGF”.
- In (B) 30 min, P-TIE2, “rh-ANG-2+rh-VEGF” is most highest expression in group. However, it does not seem to be in (A).
- Between (A) and (B), TIE2, there are many differences.
- In (D) 60 min, P-TIE2, “Ctrl” is most expression in group, However, In (C), “rh-ANG-2” is more expressed than “Ctrl”
-In (F) 60 min, P-VEGFR1, “rh-ANG-2” is more expressed than “Ctrl”, However, in (E), “rh-ANG-2” is less expressed than “Ctrl”.

Moreover, there is still differences between the bands and the graph. Please check, and make it clear.

3.      Same as previous comment, in Figure7, there are differences between the bands and the graph. Please check carefully, and make it clear.

Author Response

In the manuscript titled “ANG-2 and VEGF promote migration and invasion in hepatocellular carcinoma- and intrahepatic cholangiocarcinoma-derived spheroids”, Adriana et al investigated the regulation of HCC and iCCA-derived spheroids to migrate and invade through ANG-2 and VEGF. This manuscript is well written, with only a few comments.

  1. In Figure 2, authors should provide probability of statistical analysis with p-value.

According to the reviewer's comment we specified in the caption of Figure 2 that the p-value obtained after statistical analysis of viability assay data was not significant. Comparisons were performed among untreated versus treated spheroids for each condition, with p>0.05 in all cases.

  1. In Figure 6, there are differences between the bands and the graph. For example,

- In (A) 15 min, P-TIE2, “rh-VEGF” is more expressed than “rh-ANG-2”. However, in (B), “rh-ANG-2” shown higher expression than “rh-VEGF”.

- In (B) 30 min, P-TIE2, “rh-ANG-2+rh-VEGF” is most highest expression in group. However, it does not seem to be in (A).

- Between (A) and (B), TIE2, there are many differences.

- In (D) 60 min, P-TIE2, “Ctrl” is most expression in group, However, In (C), “rh-ANG-2” is more expressed than “Ctrl”

-In (F) 60 min, P-VEGFR1, “rh-ANG-2” is more expressed than “Ctrl”, However, in (E), “rh-ANG-2” is less expressed than “Ctrl”.

Moreover, there is still differences between the bands and the graph. Please check, and make it clear.

We acknowledge the reviewer's observation. However, there is an explanation for these apparent inconsistencies. The densitometric data presented in the graph are derived from the average of three independent experiments. In each of these experiments, we normalized the densitometric values of the marker bands to those of actin to minimize variability attributable to actin content. This standardization accounts for the instances where the columns in the graph may not seem to correspond directly with the visualized bands.

For clarity, we report the raw actin band data presented in Figure 6 for the sample cited by the reviewer. 

- In Figure 6 (A) at 15 min, the actin value for the rh-ANG-2 sample is 3.851.948, while the actin value for the rh-VEGF sample is 4.673.118. As a result, the P-TIE2 column in the graph is higher in the rh-ANG-2 sample compared to the rh-VEGF sample.

- In Figure 6 (B) at 30 min the actin levels were as follows:  Ctrl 4.132.912; rh-ANG-2 3.644.595; rh-VEGF 4.047.246; rh-ANG-2+rh-VEGF 3.772.100 thus determining the same apparent discrepancy  in the graph.

- In Figure (D) at 60 min the raw data for actin were: Ctrl 4.952.340 and rh-ANG-2 is 5.614.700.

- In (F) 60 again the different actin levels at densitometry for  Ctrl (3.105.729) and rh-ANG-2 (1.545.140) explain the differences in the graphical representation of the bands.   

  1. Same as previous comment, in Figure7, there are differences between the bands and the graph. Please check carefully, and make it clear.

As for the previous comment, the explanation resides in the normalization process with actin. We have previously uploaded the original blots both for marker and for the actin that supports our explanation. Moreover, in the previous revision of the manuscript, we had already replaced some blots, whenever possible, with some that more accurately reflected the densitometry results.

Reviewer 2 Report (Previous Reviewer 2)

Comments and Suggestions for Authors

The manuscript has been significantly enhanced. I have the following minor/ major comments. 

-In the abstract mention that Hep3B and HepG2 are HCC

-Notably, approximately 20% of the iCCA biopsies we analyzed for gene expression 9 exhibited the TS (data not shown).= add this data as supplementary figure or Table.

-Studies by Dong et al. [24], Ribatti et al. [25], and Giannelli et al. [26] have shown that EMT 47 plays a key role in driving tumor progression by increasing invasion and metastasis.= These are not the most relevant references. Please use the most related references on EMT and tumor growth and metastasis. For example, refer to the introduction and discussion of  https://pubmed.ncbi.nlm.nih.gov/30445167/    DOI: 10.1016/j.cellsig.2018.11.007     

-Figure 3A, rh_VGFA + rhANG-2 treatment does not have a synergistic effect. Comment on this in the discussion. Same for invasion results in Figure 5.

-Figure 4A, Compare the spheroids of Control and Control+T= it seems that T does not inhibit un-stimulated migration, on the contrary the Control+ T spheroids seem to have migrated more? Comment on this in the results and discussion or add a better representative image. 

- Given the relation of inflammation to HCC and not CCA, the pro- and  anti-inflammatory roles of both of VEGF-2 and bevacizumab need to be included in the discussion. For example, bevacizumab in breast cancer, Bevacizumab was shown to induce inflammation  https://pubmed.ncbi.nlm.nih.gov/30445167/    DOI: 10.1016/j.cellsig.2018.11.007

The same has been reported  for intraocular inflammation following intravitreal bevacizumab injection   doi: 10.1136/bjo.2009.166033  . Other studies reported that Bevacizumab alleviates inflammation doi:  10.3389/fphar.2018.00649 https://www.frontiersin.org/articles/10.3389/fphar.2018.00649/full    .

For HCC, VEGFA (https://www.ncbi.nlm.nih.gov/pmc/articles/PMC9930434/ and https://www.ncbi.nlm.nih.gov/pmc/articles/PMC7139884/ )

VEGFA and  its receptor (VEGFR) have  immunosuppressive effects such as  inhibition of dendritic cell maturation, thereby allowing  immune suppressive cell infiltration and increase of the expression of immune checkpoint molecules (https://pubmed.ncbi.nlm.nih.gov/31156623/ )

Minor modifications:

Line 37 page 2= rh-ANG-2= define rh again. Rh Defined in abstract, but need to be defined again in rest of manuscript. Abstract is independent from rest of manuscript. 

Line 7 page 3= This included two cell lines from liver cancer (HepG2 and Hep3B)= mention that these are HCC

Line 8= -one from intrahepatic biliary tract cancer (HuCC-T1) and one from extrahepatic biliary tract cancer (EGI-1)= -HuCC-T1 derived from an intrahepatic biliary tract cholangiocarcinoma (iCCA) and EGI-1 derived from an extrahepatic biliary tract CCA.

Line 19 page 5= The differences were considered significant for p value<0.05.= italicize p everywhere.

Line 21 page 5= To test whether rh-ANG-2 and rh-VEGF could influence the number of viable cells, we performed a viability assay for 3D culture= To test whether rh-ANG-2 and rh-VEGF could influence the number of viable HCC or CCA cells 3D culture, we performed a viability assay of HCC and CCA spheroids.

Comments on the Quality of English Language

Ok.

Author Response

The manuscript has been significantly enhanced. I have the following minor/ major comments.

  • In the abstract mention that Hep3B and HepG2 are HCC

According to the reviewer’s comment we specified HCC in the abstract.

  • Notably, approximately 20% of the iCCA biopsies we analyzed for gene expression 9 exhibited the TS (data not shown).= add this data as supplementary figure or Table.

Following the reviewer’s suggestion, we included a Figure in Supplementary materials, reporting the cases TS+ and TS-, as described in the introduction (Figure S1). 

  • Studies by Dong et al. [24], Ribatti et al. [25], and Giannelli et al. [26] have shown that EMT plays a key role in driving tumor progression by increasing invasion and metastasis.= These are not the most relevant references. Please use the most related references on EMT and tumor growth and metastasis. For example, refer to the introduction and discussion of https://pubmed.ncbi.nlm.nih.gov/30445167/    DOI: 10.1016/j.cellsig.2018.11.007    

Following the reviewer's suggestion we introduced other relevant references regarding the correlation between EMT markers and tumor growth and metastasis in in vitro studies.

  • Figure 3A, rh_VGFA + rhANG-2 treatment does not have a synergistic effect. Comment on this in the discussion. Same for invasion results in Figure 5.

We appreciate the reviewer's observation concerning the important issue of the lack of synergistic effect between ANG-2 and VEGF in rh_VEGFA + rhANG-2 in Figure 3 and 5. Despite most data in the literature report a synergy between VEGF and ANG2, few studies have reported an antagonistic effect of ANG-2 on VEGF, both in vitro and in vivo models. We have reported in the discussion this important issue (page 15 , lines 45-51). 

  • Figure 4A, Compare the spheroids of Control and Control+T= it seems that T does not inhibit un-stimulated migration, on the contrary the Control+ T spheroids seem to have migrated more? Comment on this in the results and discussion or add a better representative image.

We acknowledge the reviewer's comment on the experiment involving inhibition analysis. The reviewer correctly pointed out that the initially provided image did not effectively convey the quantification of spheroid size as a measure of cell migration, which was derived from three independent experiments. Accordingly, we have substituted the original image with a new one that accurately reflects the size of spheroids at consistent treatment intervals (3, 24, and 48 hours post-stimulation). Moreover, it is essential to highlight that in the Trebananib experiments, the significant finding is Trebananib’s robust inhibition of rh-ANG-2 stimulated migration. Notably, Trebananib does not influence migration under control conditions, suggesting that the inhibition mechanism is ANG2-dependent.

-  Given the relation of inflammation to HCC and not CCA, the pro- and  anti-inflammatory roles of both of VEGF-2 and bevacizumab need to be included in the discussion. For example, bevacizumab in breast cancer, Bevacizumab was shown to induce inflammation  https://pubmed.ncbi.nlm.nih.gov/30445167/    DOI: 10.1016/j.cellsig.2018.11.007

The same has been reported  for intraocular inflammation following intravitreal bevacizumab injection   doi: 10.1136/bjo.2009.166033  . Other studies reported that Bevacizumab alleviates inflammation doi:  10.3389/fphar.2018.00649 https://www.frontiersin.org/articles/10.3389/fphar.2018.00649/full .

For HCC, VEGFA (https://www.ncbi.nlm.nih.gov/pmc/articles/PMC9930434/ and https://www.ncbi.nlm.nih.gov/pmc/articles/PMC7139884/ ) and  its receptor (VEGFR) have  immunosuppressive effects such as  inhibition of dendritic cell maturation, thereby allowing  immune suppressive cell infiltration and increase of the expression of immune checkpoint molecules (https://pubmed.ncbi.nlm.nih.gov/31156623/ ).

We appreciate the reviewer's insights on the relationship between Bevacizumab and inflammation. While our study primarily focused on Bevacizumab's inhibitory action on VEGF, we did note its potential role in promoting migration, invasion, and inflammation in the discussion section.  We added the references indicated by  reviewer as pertinent to this section of the Discussion.  Further exploration of pro-inflammatory markers in our HCC cell models post-Bevacizumab treatment could be a valuable avenue for future research.

Minor modifications:

Line 37 page 2= rh-ANG-2= define rh again. Rh Defined in abstract, but need to be defined again in rest of manuscript. Abstract is independent from rest of manuscript.

According to the reviewer’s comment we defined Rh in the manuscript.

Line 7 page 3= This included two cell lines from liver cancer (HepG2 and Hep3B)= mention that these are HCC

Following the reviewer’ suggestion we specified that HepG2 and Hep3B are HCC.

Line 8= -one from intrahepatic biliary tract cancer (HuCC-T1) and one from extrahepatic biliary tract cancer (EGI-1)= -HuCC-T1 derived from an intrahepatic biliary tract cholangiocarcinoma (iCCA) and EGI-1 derived from an extrahepatic biliary tract CCA.

Following the reviewer’ suggestion we modified the sentence.

Line 19 page 5= The differences were considered significant for p value<0.05.= italicize p everywhere.

According to the reviewer’s comment we have replaced p value<0.05 with p value<0.05.

Line 21 page 5= To test whether rh-ANG-2 and rh-VEGF could influence the number of viable cells, we performed a viability assay for 3D culture= To test whether rh-ANG-2 and rh-VEGF could influence the number of viable HCC or CCA cells 3D culture, we performed a viability assay of HCC and CCA spheroids.

Following the reviewer’ suggestion we modified the sentence.

Reviewer 3 Report (Previous Reviewer 3)

Comments and Suggestions for Authors

The authors have addressed the reviewer's comments.

Author Response

We thank Reviewer #3 for approving our previous response to his/her/their comments

Round 2

Reviewer 2 Report (Previous Reviewer 2)

Comments and Suggestions for Authors

The authors addressed my comments.

Minor suggestion to correct the recently added statement.

"Similar to the migration assay, combining rh-ANG-2 and rh-VEGF did not 32 produce a discernible effect in the invasion assays for either Hep3B or HuCC-T1 cells. This 33 lack of effect implies that these factors might act as antagonists in our tumor models.

The ANG-2 and VEGF effects could be antagonsitic or additive in this study. There is not enough evidence to point out any of these.

Comments on the Quality of English Language

English is appropriate.

Minor typos.

This manuscript is a resubmission of an earlier submission. The following is a list of the peer review reports and author responses from that submission.

Round 1

Reviewer 1 Report

Comments and Suggestions for Authors

In the manuscript titled “Ang-2 and VEGF promote migration and invasion in hepatocellular carcinoma- and intrahepatic cholangiocarcinoma-derived spheroids”, the authors investigated whether Ang-2 and VEGF could promote migration and invasion activity in HCC. Overall, the manuscript suggests what the authors want to argue. However, there are some improvements needed to improve the current version of the manuscript. Here are several comments that need to be addressed,

# Comment 1.

There is too much space in the title.

# Comment 2.

Genes and proteins must be distinguished. Write genes’ name in italics.

Ex) E-cadherin (CDH1), N-cadherin (CDH2) and Vimentin (VIM)

# Comment 3.

In the introduction section, there is a lack of information about the clinical impact of EMT on HCC or iCCA. Reinforce the introduction part.

# Comment 4.

The scale bar representation in Figure 1 is blurry. Change to a clear image.

# Comment 5.

It is recommended to write the number of cells as 2  103 or 2,000. Correct all sentences expressing cell count.

# Comment 6.

Same words must be expressed the same way. EX) HepG-2 vs. HepG2, Ang-2 vs. Ang2

# Comment 7.

In the Figure 2 legends, check whether the the numeric expression in this part [(HepG2 3,6 x 106; Hep3B 3,8 x 106; HuCCT-1 9,6 x 105; EGI-1 2,4 x 106)] is a comma or a point. Also replace x with the multiplication expression .

EX) HepG2 3,6  106 ­­­vs. HepG2 3.6  106

# Comment 8.

The authors treated rh-Ang-2 and rh-VEGF for 3 hours and 48 hours. Why do they choose 3 or 48 h? Please specify the reason for the times.

# Comment 9.

The Results section must be specified and explained in alphabetical order of the figure. Modify it so that it is (Figure 4A, B), not (Figure 4A, E). Check and modify the overall results part, especially the results section depicting Figure 4, 5, and 6.

# Comment 10.

The sizes of the scale bars are all different. For accurate comparison, it is required to have the same photo size for all four cell lines. Ex) Figure 4 Figure 5

# Comment 11.

References need to be added to both the introduction and discussion parts.

# Comment 12.

Overall, the descriptions of the Figures in the Results section were not well organized. It must be well organized.

Reviewer 2 Report

Comments and Suggestions for Authors

The manuscript has scientific merit, but I find major flaws in experimental design and choice of cell lines. The authors need to attend to the following

1.     In fact the logic of the experiments needs to be thought out. In the environment of a hepatic cancer, the cancer cells would secrete ANG-2 and VEGFA. The secreted ANG-2 and VEGFA would act on endothelial cells and other cells of the cancer microenvironment. Therefore, the major effect of VEGF and ANG-2 are paracrine, rather than autocrine. Autocrine signaling of ANG-2 and VEGFA in HCC and iCCA in vivo, may exist, but the authors need to list evidence of this autocrine signaling and mention that they are testing this signaling. The cell culture model used by the authors cannot test paracrine signaling of VEGFA or ANG-2.

2.     The authors did not justify their choice of the HCC and iCCA cell lines they have selected. The authors affirm that aggressiveness of HCC and iCCA is correlated with overexpression of ANG-2. Therefore, the authors should compare ANG-2 and VEGF protein levels in different HCC and iCCA cell lines and choose the cell lines with highest ANG-2 +/- high VEGF expression. The authors can also chose HCC and iCCA cell lines with high and low ANG-2 protein levels and compare their migration and invasion properties. This is the first preliminary experiment to be performed. Otherwise, the authors can test the levels of ANG-2 and VEGF in the cell lines they have used in the study in comparison to “Normal” hepatic cell lines. In this case they need to show that ANG-2 is upregulated in these cell lines in comparison to the “Normal” hepatic cells. This is needed in light of the hypothesis set by the authors. In fact, Figure 3 shows that these cell lines have low levels of ANGP-2 and VEGFA. Therefore, the authors need to use other cell lines!

In addition, Low levels of ANG-2 in these cell lines may justify the lack of effect on proliferation of these cell lines in Figure 2. Lack of effect on cell proliferation needs to be discussed in more detail in the discussion.

3.     The authors need also to check the levels of the receptors of ANG-2 and VEGFA. Also, checking the activation of these receptors is needed; by checking receptor phosphorylation or phosphorylated downstream targets.

4.     Since ANG-2 and VEGFA are secreted proteins, it is suggested to test their levels in the media of the cells by ELISA and not in the total cell lysates by Western blotting.

5.     In Figure 3, the authors say that the levels of VEGFA and ANG-2 are low. Low compared to what? The authors need to compare expression levels relative to “Normal” hepatic cell lines or primary liver cells.

6.     “HCC has a strong association with the inflammatory background caused by chronic liver disease, while iCCA has a weaker relationship with inflammation”. Why the authors did not test inflammatory mediators in cells in culture? Discuss inflammation and its implications for cancer aggressiveness. Discuss this in the discussion, in light of the fact that Bevacizumab may cause inflammation and this may contribute to resistance to anti-angiogenesis therapies. This has been shown in breast cancer “Bevacizumab induces inflammation in MDA-MB-231 breast cancer cell line and in a mouse model  doi: 10.1016/j.cellsig.2018.11.007”, use this reference to discuss in relation to hepatic cancers.

7.     The authors need to introduce the difference of treatment with ANG-2 versus VEGF, in the introduction. Also, need to mention if there is a crosstalk between ANG-2 and VEGFA . During this discussion, justify the use of combination treatment.

8.     Use proper nomenclature of human genes and proteins, also distinguish between human and mouse genes and proteins. Check MDPI policy on abbreviations used for human and mouse genes and proteins= for example Ang-2 protein should be ANG-2 protein, and Ang-2 is the mouse protein, etc…

9.     Many language errors exist. The manuscript needs to be proof-read by a native English speaker.

Examples of typos follow.

- In Abstract,  share 95% of up-regulated genes= change share into shares

- and similar severe prognosis with intrahepatic cholangiocarcinoma (iCCA) proliferative subgroup= and a similar poor prognosis as the proliferative subgroup of intrahepatic cholangiocarcinoma (iCCA).

- What does bland mean, in bland HCC?

- Prognosis is more severe and resistance to therapy higher in iCCA= Prognosis is more poor and resistance to therapy is higher in iCCA

- which can be found also in about 20 % of iCCA tested= what does iCCA tested mean? Do you mean = of the tested iCCA= tested for what and in what context?

- et al. not et al= correct everywhere

- Two thousand (HCC cell lines) or 1500 (CCA cell lines)= Two thousand HCC cells/ well (of a 6 or 12 or 24-cell culture plate) or 1500 cells/well of CCA were seeded

- or with 100 ng/mL of each protein combined= or with a combination of 100 ng/mL of each protein

- we started the observation 3 h after stimulation and additional time points after 24 and 48 h. What does this mean? 24 and 48 h are after 3 h? correct.

Minor:

1.     Under pathological conditions, Ang-2 is responsible for neoangiogenesis= this is under both physiological and pathological conditions? = check and correct. Seaprate into two sentences. Write a separate sentence about role in angiogenesis, and then another sentence about role in cancer..

2.     were transferred into attachment plates= specify what is meant by attachment plates?

3.     Migration and invasion assay Fifty-thousand cells/well were seeded= specify which kind of well? The same for “Four hundred thousand cells/well were “= Specify type of well everywhere.

4.     In some experiments, inhibitors were added after stimulation with rh-Ang-2 or rh-VEGF= specify the exact time of addition of the inhibitors after treatment? Did the authors try pre-incubation with inhibitors?

Comments on the Quality of English Language

Many language errors exist. The manuscript needs to be proof-read by a native English speaker.

Examples of typos follow.

- In Abstract,  share 95% of up-regulated genes= change share into shares

- and similar severe prognosis with intrahepatic cholangiocarcinoma (iCCA) proliferative subgroup= and a similar poor prognosis as the proliferative subgroup of intrahepatic cholangiocarcinoma (iCCA).

- What does bland mean, in bland HCC?

- Prognosis is more severe and resistance to therapy higher in iCCA= Prognosis is more poor and resistance to therapy is higher in iCCA

- which can be found also in about 20 % of iCCA tested= what does iCCA tested mean? Do you mean = of the tested iCCA= tested for what and in what context?

- et al. not et al= correct everywhere

- Two thousand (HCC cell lines) or 1500 (CCA cell lines)= Two thousand HCC cells/ well (of a 6 or 12 or 24-cell culture plate) or 1500 cells/well of CCA were seeded

- or with 100 ng/mL of each protein combined= or with a combination of 100 ng/mL of each protein

- we started the observation 3 h after stimulation and additional time points after 24 and 48 h. What does this mean? 24 and 48 h are after 3 h? correct.

Reviewer 3 Report

Comments and Suggestions for Authors

The scientific writing and data presentation needs to be largely improved for the convenience of the readers.

1. Scale bar should be clearly demonstrated in all images.

2. Was there statistical analysis in the results of figure 2?

3. The style of bar figures should be reorganized throughout the manuscript.

4. In figure 3, 5 and 8, the trend in the protein expression in images of western blot analysis was not consistent with that of the quantitative results.

5. The style pf data presentation in figure 4, 5 and 8 was not friendly for the readers.

Comments on the Quality of English Language

Moderate editing of English language required